# Catch Your Emotion: Sharpening Emotion Perception in Multimodal Large Language Models

Yiyang Fang [* 1]  Jian Liang [* 1]  Wenke Huang [* 1]  He Li [1]  Kehua Su [1]  Mang Ye [1 2]

## Abstract

Multimodal large language models (MLLMs) have achieved impressive progress in tasks such as visual question answering and visual understanding, but they still face significant challenges in emotional reasoning. Current methods to enhance emotional understanding typically rely on fine-tuning or manual annotations, which are resource-intensive and limit scalability. In this work, we focus on improving the ability of MLLMs to capture emotions during the inference phase. Specifically, MLLMs encounter two main issues in the inference stage: they struggle to distinguish between semantically similar emotions, leading to misclassification, and they are overwhelmed by redundant or irrelevant visual information, which distracts from key emotional cues. To address these, we propose a training-free method named Sharpening Emotion Perception in MLLMs (SEPM), which incorporates a Confidence-Guided Coarse-to-Fine Inference framework to refine emotion classification by guiding the model through simpler tasks. Additionally, SEPM employs Focus-on-Emotion Visual Augmentation to reduce visual redundancy by directing the attention of models to relevant emotional cues in images. Experimental results demonstrate that SEPM significantly improves MLLM performance on emotion-related tasks, providing a resource-efficient and scalable solution for emotion recognition. Our code is available in *https://github.com/fuyyyyy/SEPM*.

---
[*]Equal contribution  [1]National Engineering Research Center for Multimedia Software, School of Computer Science, Wuhan University, Wuhan, China [2]Taikang Center for Life and Medical Sciences, Wuhan University, Wuhan, China. Correspondence to: Kehua Su <skh@whu.edu.cn>, Mang Ye <yemang@whu.edu.cn>.

*Proceedings of the $42^{nd}$ International Conference on Machine Learning*, Vancouver, Canada. PMLR 267, 2025. Copyright 2025 by the author(s).

## 1. Introduction

Multimodal large language models (MLLMs) have gained increasing attention in various fields due to their powerful understanding and reasoning capabilities (Liu et al., 2023; Chen et al., 2024; Li et al., 2024b; Wang et al., 2024; Li et al., 2024a). In recent years, driven by the rapid advancement of deep learning (Huang et al., 2024; 2023), they have shown exceptional performance in traditional tasks such as visual question answering (Lee et al., 2024), visual understanding and reasoning (Zhang et al., 2025). When it comes to the emotional domain, advanced MLLMs seem to lack the ability to accurately interpret and respond to emotional cues (Yang et al., 2024a). Although current MLLMs can generate basic emotional responses, they still struggle to fully understand and capture the complexity of emotions.

Recently, some studies have attempted to address the shortcomings of MLLMs in capturing emotional cues (Xing et al., 2024; Yang et al., 2024a; Zhao et al., 2024a; Zhang et al., 2024a). These efforts can mainly be divided into fine-tuning methods and training-free methods. Regarding the former, different encoders are often introduced to interpret emotional information across various modalities (Cheng et al., 2024), or an additional MLLM (Achiam et al., 2023) is employed to generate corresponding textual descriptions (Xie et al., 2024). Fine-tuning on emotion-related datasets is then carried out to enhance the model's ability to understand emotions. The additional fine-tuning in these methods incurs significant training costs, which can limit their scalability and practicality in applications. Therefore, training-free methods serve as an alternative approach to enhance the ability of MLLMs to capture emotions. Some studies guide MLLMs to pay more attention to emotion-related regions through visual prompting (Zhang et al., 2024b), but such methods require prior annotation of images, resulting in additional overhead. Building on this, we aim to enhance the ability of MLLMs to capture emotions solely during the inference stage, without relying on additional training or manual annotation. This not only significantly reduces resource consumption but also improves the efficiency and flexibility of the model in real-world applications.

In the inference stage, MLLMs face unique challenges when handling emotion classification tasks. On the one hand,

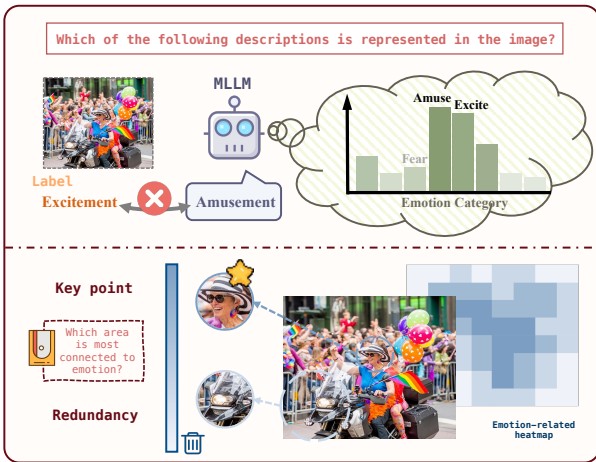

*Figure 1.* Illustration of the challenges faced by Multimodal Large Language models (MLLMs) in emotion tasks. The upper half of the figure demonstrates that semantically **similar emotion categories** are more prone to confused inference than distinctly different ones. The lower half points out that **different patches of the image** have varying importance when conveying emotion information, and some redundant information is present.

since MLLMs perform classification tasks based on category word meanings, ❶ *semantically similar emotion categories often lead to confusion during inference*, as the model struggles to differentiate between subtle variations in emotional cues. For example, distinguishing between "amusement" and "excitement" can be challenging for the model due to their similar emotional cues, as illustrated in Fig. 1. Without an accurate understanding of context, the model might misclassify one as the other. On the other hand, ❷ *the visual information conveyed when expressing emotions contains a certain degree of redundancy*, which can prevent the model from effectively extracting key emotional cues. As Fig. 1 illustrates, emotional information is often concentrated in specific local regions of an image. It's crucial to direct the model's focus to these regions and reduce irrelevant background noise. Effectively leveraging more reliable textual input (Yang et al., 2024b) in combination with these visual cues can significantly enhance the model's ability to capture and interpret emotions accurately.

In this paper, we propose Sharpening Emotion Perception in Multimodal Large Language Models (SEPM), which consists of two primary components. **First**, due to ❶, we design a two-stage inference framework named Confidence-Guided Coarse-to-Fine Inference. The first stage performs coarse-grained emotion classification, generating textual prompts for the next stage based on the results of positive or negative emotion classification and their confidence levels. The second stage then completes fine-grained emotion classification based on this enhanced input. This approach allows the MLLM to focus on a relatively simpler task at each step, thereby improving the model's inference accuracy. **Second**, we introduce Focus-on-Emotion Visual Augmentation to

address ❷. We guide the MLLM to focus on emotional information through Focus on Emotion (FoE) textual prompts. The relevance of visual tokens to FoE is calculated based on the attention map obtained during the first inference. By dropping visual tokens unrelated to FoE, we reduce visual redundancy and enable the model to better capture and interpret emotional cues. Experimental results show that our approach significantly improves MLLM performance on emotion-related tasks without requiring additional training. The main contributions are as follows:

- To address inference confusion from semantically similar emotions, we propose Confidence-Guided Coarse-to-Fine Inference. The two-stage inference allows the model to focus on relatively simpler tasks at each stage, thereby improving inference accuracy.

- We propose Focus-on-Emotion Visual Augmentation to address redundant emotional information in images. We eliminate visual tokens unrelated to the Focus-on-Emotion prompt, improving the ability of MLLMs to detect and interpret emotions.

- We conduct experiments on various emotion datasets to demonstrate that our method improves the performance of MLLMs on emotion tasks, without requiring additional training or manual annotations.

## 2. Related work

### 2.1. Multimodal Large Language Models

Multimodal Large Language Models (MLLMs) (Liu et al., 2023; Chen et al., 2024; Li et al., 2024b; Wang et al., 2024; Li et al., 2024a) have garnered significant attention for their strong reasoning capabilities (Liu et al., 2025b; Jin et al., 2025; 2024) and ability to handle a variety of tasks across different modalities (Huang et al., 2025a;b; Liang et al., 2025). Most research in this field focuses on leveraging pre-trained models for general applications (Liu et al., 2024a; Zhao et al., 2024b; Han et al., 2024; Bai et al., 2025), including vision-language tasks such as multimodal reasoning (Lu et al., 2022; Masry et al., 2022; Wang et al., 2025), image captioning (Lin et al., 2014; Young et al., 2014), visual question answering (Hudson & Manning, 2019; Goyal et al., 2017; Singh et al., 2019; Huang et al., 2025c). MLLMs have demonstrated excellent performance in these general tasks (Bi et al., 2024), showcasing their ability to integrate and reason across multiple modalities.

However, MLLMs often struggle with emotion-related tasks (Lian et al., 2025; Fang et al., 2025; Yang et al., 2025). To address this, several studies have focused on fine-tuning MLLMs using emotional datasets (Xing et al., 2024; Yang et al., 2024a; Zhao et al., 2024a; Zhang et al., 2024a). EmoVIT (Xie et al., 2024) leverages GPT-4 to generate

relevant textual descriptions, aiding models in better understanding emotions and improving their ability to interpret emotional cues. Meanwhile, Emotion-LLaMA (Cheng et al., 2024) integrates specialized encoders that are specifically designed to capture and interpret emotional information across different modalities, thereby enhancing its capacity to understand and reason about emotions from diverse inputs, including text, audio, and visual cues. These methods require fine-tuning, leading to high training costs. We aim to enhance the emotional recognition capabilities of MLLMs during inference, without relying on extra training, making emotional understanding more efficient and scalable.

### 2.2. Training-Free in MLLMs

Due to the high training costs associated with MLLMs, training-free methods have gained considerable attention as a fresh perspective (Zhu et al., 2024; Ding et al., 2024; Bi et al., 2025), particularly in tasks such as inference acceleration (Chen et al., 2025; Zhang et al., 2024c; Bolya et al., 2022; Choudhury et al., 2024) and mitigating MLLM hallucinations (Kim et al., 2024b; Liu et al., 2025a; Kim et al., 2024a). Typical training-free methods involve techniques such as text prompt engineering to guide model behavior and improve outcomes without additional training, token merging to enhance processing efficiency by simplifying input data, and visual augmentation alongside multi-step inference to boost performance through visual enhancements and iterative processing. In emotion-related tasks, Zhang et al. (Zhang et al., 2024b) propose Set-of-Vision (SoV), a method for identifying the locations of faces in images, thereby enhancing facial expression recognition in MLLMs. However, this approach brings the additional cost of image annotation and is limited to tasks that focus solely on facial features. Thus, we strive to develop a training-free method to eliminate the need for additional data modifications while effectively addressing a wider range of emotional cues.

## 3. The Proposed Method

### 3.1. Preliminary

The architecture of a multimodal large language model (MLLM) typically includes three components: a visual encoder $f$, such as ViT, an LLM $g$, such as Vicuna and LLaMA, and a connector module $\varphi$. For a query instance, the input consists of a visual image $x_v$ and a textual instruction $x_t$. The corresponding label is the language response $y$. Initially, visual features $z^v$ are extracted using $f(x^v)$. Subsequently, the trainable projection $\varphi$ is applied to transform $z^v$ into language embedding tokens $h^v = \varphi \cdot z^v$. Meanwhile, the textual instruction is tokenized as $h^t = \text{Tokenize}(x^t)$. The visual and textual tokens are then combined and input into the LLM module $g$, which generates the language output $\hat{y} = g([h^v, h^t])$. In our work, we enhance the emotion-

capturing capability of MLLMs by optimizing the inference process, eliminating the need for additional training.

### 3.2. Confidence-Guided Coarse-to-Fine Inference (CCI)

**Motivation.** Considering the large number of emotion categories, MLLMs performing emotion classification are prone to confusion due to the influence of semantically similar emotion categories. However, requiring MLLMs to distinguish between positive and negative emotions is significantly easier. Further enhancing emotion classification by using positive and negative emotions as prompts can lead to higher inference accuracy. Therefore, we design a two-stage inference process following a Coarse-to-Fine approach, which enables the model to concentrate on simpler tasks during inference.

**Coarse-Grained Stage.** We pre-divide all emotion categories into the positive emotion categories (PEC) and the negative emotion categories (NEC), and generate the following Coarse-grained Query ($\mathcal{Q}_c$).

---

**Coarse-grained Query**:
*Which of the following descriptions best represents the image?*

      **A.** Positive      **B.** Negative

*Positive emotions include* [PEC]. *Negative emotions include* [NEC]. *Answer directly with the letter of the chosen option.*

---

Then, we input this query along with the sample $D$ into the MLLM for inference:

$$\hat{\mathcal{E}} = \mathcal{M}(\mathcal{Q}_c, D), \tag{1}$$

where $\hat{\mathcal{E}}$ represents the positive or negative emotion, and $\mathcal{M}$ represents the inference function of the MLLM.

**Fine-Grained Stage.** Based on the positive and negative emotion label $\hat{\mathcal{E}}$ obtained from the coarse-grained stage, we modify Fine-grained Queries ($\mathcal{Q}_f$) for different samples, to simplify the classification task.

---

**Fine-grained Query**:
*Which of the following descriptions best represents the image?*

      **[PEC] / [NEC]**

*Answer directly with the letter of the chosen option.*

---

For samples with positive emotions, we retain only the options from the positive emotion categories, while for negative emotions, we keep the negative ones. Similarly, we feed sample $D$ and its corresponding query into the MLLM, which results in the specific emotion category $\mathcal{E}$:

$$\mathcal{E} = \mathcal{M}(\mathcal{Q}_f, D). \tag{2}$$

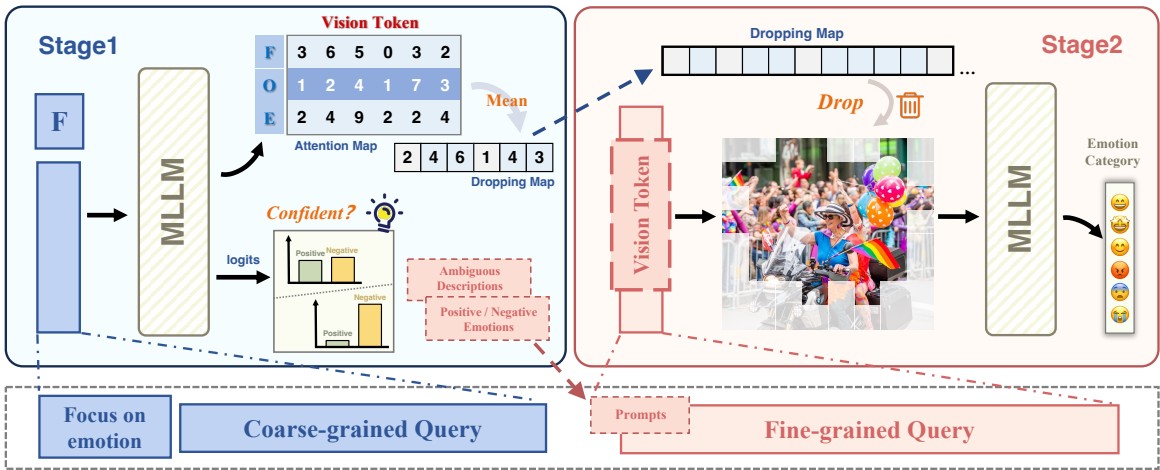

*Figure 2.* **Architecture illustration** of SEPM involves two main stages. **Stage 1** performs coarse-grained emotion inference. The model applies the "Focus-on-Emotion" prompt optimization to establish the relationship between the visual token and emotions. Using the positive/negative emotions inferred by MLLM and the confidence calculated based on the model's logits output, the model refines and customizes the prompt optimization for the fine-grained stage. In **Stage 2**, fine-grained emotion inference is performed. The model discards redundant information based on the importance of the visual token from the coarse-grained stage and leverages the optimized prompt to achieve more precise emotion classification.

**Confidence Assessment.** *What if the inference in the coarse-grained stage is incorrect?* Due to the tendency of MLLMs to prefer textual information, biases in the coarse-grained stage can directly lead to errors in the final inference result. Thus, mitigating the impact of mistakes in the coarse-grained stage on model inference is a problem that needs to be urgently addressed.

We obtain the logits $z$ before the model output in the coarse-grained stage and compute its probability distribution.

$$z = \mathcal{M}_{logits}(\mathcal{Q}_c, D),$$
$$p = softmax(z), \tag{3}$$

where $\mathcal{M}_{logits}$ represents the logits generator, and $p$ denotes the output probability distribution. Since what we obtain is a probability distribution over the vocabulary, we find the indices corresponding to the letters A and B and calculate the probability variance between the two options:

$$\mathcal{C} = \frac{(p_A - \mu)^2 + (p_B - \mu)^2}{2}, \tag{4}$$

where $\mathcal{C}$ is the confidence score, $\mu$ is the mean, and $p_A, p_B$ are the probabilities corresponding to letter A and B.

The confidence score indicates that the smaller the value of $\mathcal{C}$, the closer the probabilities of the two options are. This suggests that the model lacks strong certainty during the reasoning process. Therefore, we set a threshold, denoted as $\alpha$, where results with a confidence score lower than $\alpha$ are considered unreliable samples. For these, we retain all emotion categories in the fine-grained stage and add **ambiguous-related descriptions** reflecting the behavior from the first stage, enhancing the prompting effect.

### 3.3. Focus-on-Emotion Visual Augmentation

**Motivation.** Generally, MLLMs tend to prioritize textual information over other modalities. To mitigate the redundancy in visual information that causes the model to overlook key emotional cues, we employ a text-based prompt to guide the optimization of visual augmentation (Liu et al., 2024b), enabling the model to focus more effectively on the most critical emotional cues.

**Focus-on-Emotion Prompts (FP).** To improve the focus of MLLMs on emotional information, We introduce a straightforward yet effective prompting technique called the Focus on Emotion (FoE) prompt, which explicitly directs the MLLM to prioritize emotional cues:

> ***Please focus on emotion.*** [Original Query]

This method can seamlessly integrate into the current model queries, as a criterion for selecting visual tokens in subsequent steps. It is highly efficient and convenient.

**Estimation of Visual Token Significance.** Subsequently, we need to understand the degree of relevance between visual tokens and emotions in order to identify which visual tokens are redundant or interfering. Therefore, it is natural to leverage the attention map ($\mathcal{A}$) from the VLM transformation layers of *the first stage* as a reference, since they already encapsulate the query results from language to vision:

$$\mathcal{A} = \mathcal{M}_{attn}(\mathcal{Q}_c, D), \tag{5}$$

$\mathcal{M}_{attn}$ is the function that computes the attention map in the final layer of attention in the MLLM. Then we extract the FoE prompt set as emotion-related text, leveraging the query-

**Algorithm 1** SEPM

---

**Input:** Multimodal Large Language Models $\mathcal{M}$, Coarse-Grained Query $\mathcal{Q}_c$, Sample $D$.
**Output:** Specific emotion category $\mathcal{E}$.

/* Stage 1:  Coarse-Grained Inference */

$\hat{\mathcal{E}} \leftarrow \mathcal{M}(\mathcal{Q}_c, D)$        ▷ Eq.(1)
$\mathcal{A} \leftarrow \mathcal{M}_{attn}(\mathcal{Q}_c, D)$        ▷ Eq.(5)
$p_A, p_B \leftarrow \text{softmax}(\mathcal{M}_{logits}(\mathcal{Q}_c, D))$        ▷ Eq.(3)

/* Confidence Assessment */

$\mathcal{C} \leftarrow [(p_A - \mu)^2 + (p_B - \mu)^2]/2$        ▷ Eq.(4)
**if** $\mathcal{C} > \alpha$ **then**
     Fine-Grained Query $\mathcal{Q}_f =$
     $\begin{cases} \mathcal{Q}_f \leftarrow \mathcal{Q}_{\text{PEC}}, & \hat{\mathcal{E}} = \text{Positive}, \\ \mathcal{Q}_f \leftarrow \mathcal{Q}_{\text{NEC}}, & \hat{\mathcal{E}} = \text{Negative}. \end{cases}$
**else**
     $\mathcal{Q}_f \leftarrow \mathcal{Q}_{\text{ambi}}$
**end**

/* Stage 2:  Fine-Grained Inference */

**for** $j = 1, 2, \ldots, L_v$ **do**
     /* Emotion-Related Sig. Estimation */
     $P[i, j] \leftarrow \mathcal{A}(i, j)$        ▷ Eq.(6)
     $\bar{P}[j] \leftarrow \frac{1}{L_t} \sum_{i=1}^{L_t} P[i, j]$        ▷ Eq.(8)
**end**
$k \leftarrow \lfloor \beta N_v \rfloor, \quad \mathcal{R} \leftarrow \text{argmin}_k(\bar{P})$        ▷ Eq.(9)
$\mathcal{V}' \leftarrow \{v_j \mid j \notin \mathcal{R}, j \in \{1, 2, \ldots, N_v\}\}$        ▷ Eq.(10)
$\mathcal{E} \leftarrow \mathcal{M}(\mathcal{Q}_f, D')$        ▷ Eq.(2)
**return** Fine-Grained Emotion Class $\mathcal{E}$.

---

dimension of textual logits and the key-dimension of the visual modality to construct the drop matrix $P \in \mathbb{R}^{L_t \times L_v}$, where $L_t$ and $L_v$ are the lengths of the text tokens (FoE) and visual tokens, respectively. The drop matrix $P$ is defined as:

$$P[i, j] = \mathcal{A}(i, j),$$
$$i \in \{x \mid \mathbb{I}[x] \in \mathbb{T}\}, \qquad (6)$$
$$j \in \{y \mid \mathbb{I}[y] \in \mathbb{V}\},$$

where $\mathbb{I}$ denotes the set of all input tokens, including both textual and visual modalities, $\mathbb{T}$ denotes FoE prompt token set and $\mathbb{V}$ is visual tokens set, i.e.

$$(\mathbb{T} \cup \mathbb{V}) \subseteq \mathbb{I}. \qquad (7)$$

To estimate the significance of the vision token, we compute the average score of all instruction tokens as:

$$\bar{P}[j] = \frac{1}{L_t} \sum_{i=1}^{L_t} P[i, j], \quad j \in \{1, 2, \ldots, N_v\}, \qquad (8)$$

where we use $\bar{P}$ as the importance indicator for visual tokens, with more significant tokens assigned higher values.

**Visual Token Augmentation (VTA).** For the importance indicator $\bar{P}$ obtained in Eq. (8), we intend for the model to focus more on the parts of the visual tokens with higher

importance. Thus, we considered that dropping redundant tokens would be an effective approach. We select the lower-ranked tokens in $\bar{P}$ to generate the dropping map $\mathcal{R}$:

$$\mathcal{R} = \text{argmin}_k(\bar{P}), \quad k = \lfloor \beta N_v \rfloor, \qquad (9)$$

where $\text{argmin}_k(\bar{P})$ represents the indices of the $k$ smallest values in $\bar{P}$. Here, $\beta$ is the dropping ratio, and $N_v$ denotes the total number of visual tokens.

For the visual input in *the second stage*, we discard redundant visual tokens based on the dropping map. The refined visual token set $\mathcal{V}'$ is defined as:

$$\mathcal{V}' = \{v_j \mid j \notin \mathcal{R}, j \in \{1, 2, \ldots, N_v\}\}, \qquad (10)$$

where $\mathcal{V} = \{v_1, v_2, \ldots, v_{N_v}\}$ represents the original set of visual tokens, with $N_v$ being the total number of tokens. The dropping map $\mathcal{R}$ contains the indices of redundant tokens determined in the first stage.

The refined visual token set $\mathcal{V}'$ and the textual prompts are combined into a new sample $D'$, which is fed into the MLLM for fine-grained inference, as in Eq. (2). This approach enhances the focus of the model on critical emotional cues, resulting in more accurate and reliable outcomes.

## 4. Experiments

In this section, we observe the performance of SEPM from multiple perspectives, and also notice several notable issues:

**RQ1:** Is there more redundant visual information in emotion-related tasks compared to general tasks? (Sec. 4.2)

**RQ2:** How should we choose which visual tokens to drop to improve inference accuracy? (Sec. 4.2, Sec. 4.3)

**RQ3:** Is it reliable to use the variance of the logits distribution output by MLLMs as a basis for confidence? (Sec. 4.3)

**RQ4:** Do MLLMs exhibit biases when performing emotion classification by different prompts? (Sec. 4.3)

### 4.1. Experimental Setup

**Architecture.** Following the multimodal large language model paradigm, we use two popular models as the foundation for our experiments to evaluate the effectiveness of our method: LLaVA (Liu et al., 2023) and VILA (Lin et al., 2024). Specifically, LLaVA uses a model with 7 billion parameters, while VILA employs a model with 8 billion parameters as the test model.

**Datasets.** We evaluate our framework on four emotion datasets, which are annotated across different scenarios and numbers of categories: Emotion6 (Peng et al., 2015), EmoSet (Yang et al., 2023), WebEmo (Panda et al., 2018), and Abstract (Machajdik & Hanbury, 2010). The WebEmo

Table 1. **Comparison** with state-of-the-art on various emotion datasets. $\beta$ is drop rate. The optimal results are denoted by **boldface**. ↑ indicates an increase in accuracy for zero-shot inference with the same architecture, while ↓ indicates a decrease. Refer to Sec. 4.2.

| Dataset | Emotion6 | EmoSet | WebEmo$_7$ | WebEmo$_{25}$ | Abstract | Average |
|---|---|---|---|---|---|---|
| Class | 6 | 8 | 7 | 25 | 8 | |
| *LLaVA-7b* | | | | | | |
| Zero-shot | 48.32 | 52.77 | 25.56 | 15.71 | 27.86 | 34.04 |
| Zero-shot-CoT | 48.65 | 51.67 | 22.93 | 15.52 | 26.43 | 33.04 |
| SparseVLM | 49.83 | 54.00 | 26.64 | 15.68 | 26.79 | 34.59 |
| SEPM ($\beta$=0.1) | **54.21**$_{\uparrow 5.89}$ | 56.04$_{\uparrow 3.27}$ | 42.39$_{\uparrow 16.83}$ | 18.26$_{\uparrow 2.55}$ | **29.29**$_{\uparrow 1.43}$ | **40.04**$_{\uparrow 6.00}$ |
| SEPM ($\beta$=0.2) | 54.04$_{\uparrow 5.72}$ | **56.24**$_{\uparrow 3.47}$ | **42.75**$_{\uparrow 17.19}$ | **18.34**$_{\uparrow 2.63}$ | 28.21$_{\uparrow 0.35}$ | 39.92$_{\uparrow 5.88}$ |
| *VILA-8b* | | | | | | |
| Zero-shot | 47.47 | 44.45 | 41.90 | 15.35 | 20.00 | 33.83 |
| Zero-shot-CoT | 44.28 | 43.20 | 41.08 | 13.57 | 21.07 | 32.64 |
| SEPM ($\beta$=0.1) | 51.35$_{\uparrow 3.88}$ | **53.11**$_{\uparrow 8.66}$ | **44.30**$_{\uparrow 2.40}$ | 17.36$_{\uparrow 2.01}$ | 25.71$_{\uparrow 5.71}$ | **38.37**$_{\uparrow 4.54}$ |
| SEPM ($\beta$=0.2) | **52.02**$_{\uparrow 4.55}$ | **53.11**$_{\uparrow 8.66}$ | 39.10$_{\downarrow 2.80}$ | **19.39**$_{\uparrow 4.04}$ | **26.43**$_{\uparrow 6.43}$ | 38.01$_{\uparrow 4.18}$ |

Table 2. **Comparison** of different dropping strategies in LLaVA. $\beta$ is drop rate. **Bold** is the best. Refer to Sec. 4.2.

| Dataset | Emotion6 | | EmoSet | |
|---|---|---|---|---|
| Drop rate | $\beta$=0.1 | $\beta$=0.2 | $\beta$=0.1 | $\beta$=0.2 |
| Random | 53.20 | 51.85 | 55.99 | 55.91 |
| Query-related | 53.54 | 53.54 | 55.99 | 56.10 |
| FoE-related | **54.21** | **54.04** | **56.04** | **56.24** |

Table 3. **Ablation study** of the key components in SEPM. **Bold** is the best. ↑ is an increase over the baseline. Refer to Sec. 4.3.

| CCI | FP | VTA | Emotion6 | EmoSet |
|---|---|---|---|---|
| | | | 48.32 | 52.77 |
| ✓ | | | 51.68$_{\uparrow 3.36}$ | 53.98$_{\uparrow 1.21}$ |
| | ✓ | | 51.52$_{\uparrow 3.20}$ | 54.28$_{\uparrow 1.51}$ |
| ✓ | ✓ | | 53.03$_{\uparrow 4.71}$ | 56.10$_{\uparrow 3.33}$ |
| ✓ | ✓ | ✓ | **54.04**$_{\uparrow 5.72}$ | **56.24**$_{\uparrow 3.47}$ |

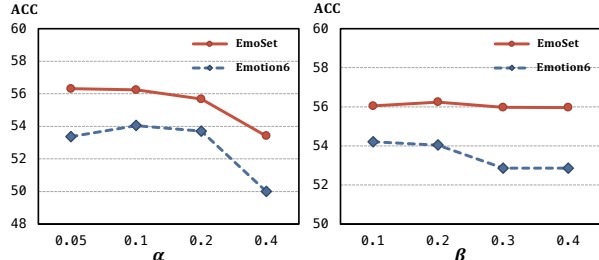

Figure 3. **Sensitivity analysis**. The results are obtained by varying the value of the corresponding hyperparameter, while fixing the others to the values adopted in the experiments. Refer to Sec. 4.3.

dataset includes emotion annotations at two levels of granularity, with 7 categories and 25 categories.

**Implement Details.** We utilize checkpoints from model pre-training and follow the template provided by the LLaVA model to design basic question prompts for inference **without further training**. To evaluate the performance of MLLMs in solving emotion-related tasks, we use answer accuracy (ACC) as the evaluation metric for our experiments. The confidence threshold $\alpha$ and drop rate $\beta$ are set to 0.1 and 0.2 by default, respectively. All experiments are conducted on 8 NVIDIA 4090 GPUs, each with 24GB of memory.

## 4.2. Comparison Experiments

We conduct comparisons on multiple emotion datasets with MLLMs Zero-shot, Zero-shot-CoT (Kojima et al., 2022) and sparseVLM (Zhang et al., 2024c). We adjust the Zero-shot-CoT prompt to better suit our task: *"Let's think step by step, then answer with the option's letter from the given choices directly."* We also evaluate different dropping strategies.

**Comparison with State-of-the-art.** We compare SEPM with the zero-shot method of MLLMs with the same architecture, as well as other training-free methods on MLLMs on various emotion datasets, as shown in Tab. 1. Firstly, Zero-

shot-CoT designs *"let's think step by step"* as a prompt enhancement. However, compared to MLLM zero-shot, the overall performance in emotion classification decreases. This suggests that simple prompt engineering does not effectively enhance the ability of MLLMs to capture emotions. Next, SparseVLM primarily improves inference efficiency by discarding task-irrelevant visual tokens. Since it is similar to our approach, we also include it for comparison here. We are pleasantly surprised to find that SparseVLM shows a slight improvement in overall performance on emotion tasks compared to zero-shot, which is in stark contrast to the general decrease in reasoning accuracy it typically experiences on more general tasks. This suggests that, compared to general tasks, MLLMs contain more redundant tokens in visual information when capturing emotions (**RQ1**). It also supports the rationale of our approach, where dropping redundant tokens helps highlight key information.

Finally, we compare SEPM under two different drop rates with previous methods and find that our approach outperforms all others on each dataset. Specifically, under the LLaVA architecture, SEPM with a 20% drop rate improve by 17.19 compared to zero-shot, and it also shows significant improvements in other comparison groups. This indicates that SEPM greatly improves the emotion-capturing ability of MLLMs without requiring any training.

**Comparison of Different Dropping Strategies.** We evaluate three different dropping strategies under the LLaVA

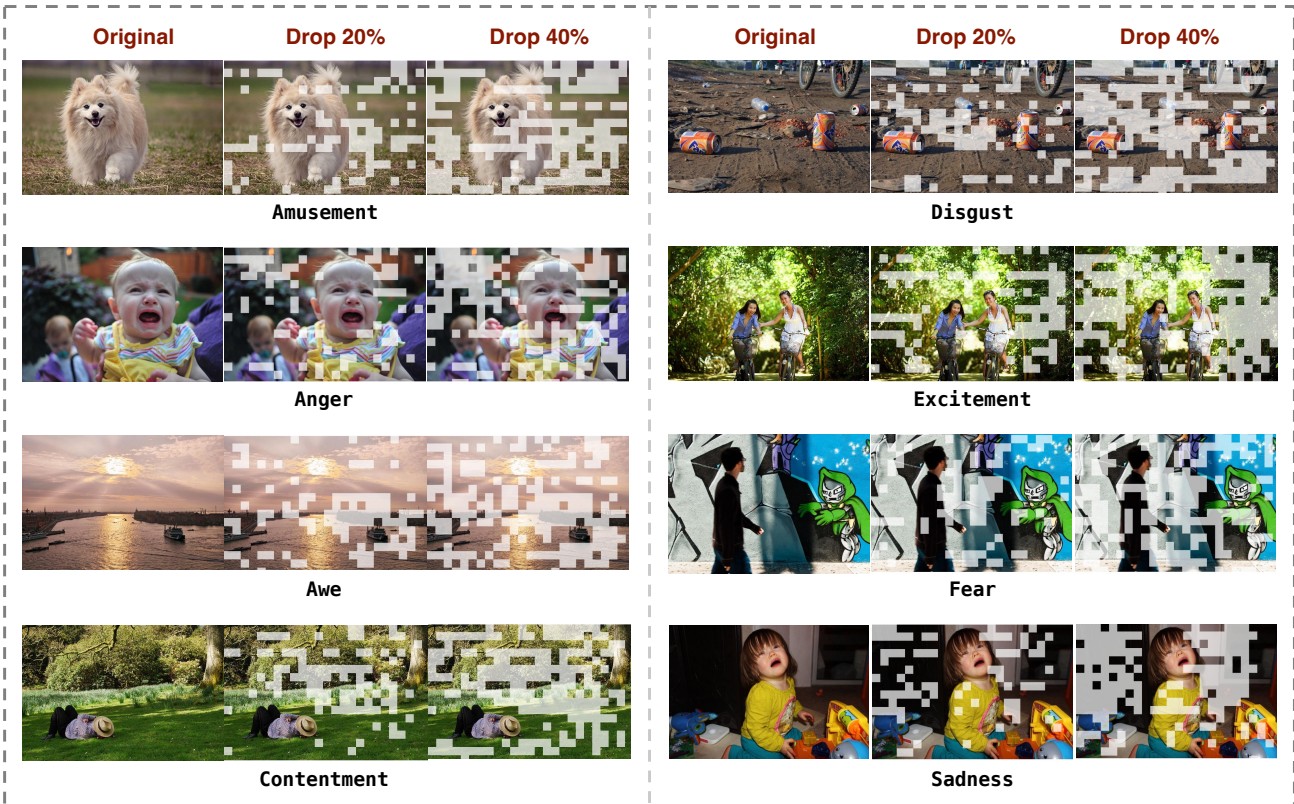

*Figure 4.* **Visualization** of dropped visual tokens for different emotions. The occluded part represents dropped tokens. Refer to Sec. 4.3.

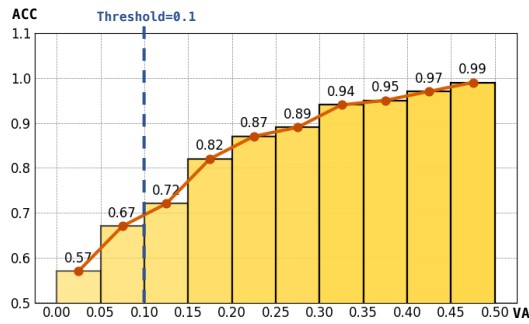

*Figure 5.* **Visualization** of confidence estimation. VAR is variance, representing the confidence level of coarse-grained inference. Threshold is our default confidence threshold $\alpha$. Refer to Sec. 4.3.

framework on the Emotion6 and EmoSet datasets. The first strategy is Random Drop, where the model randomly drops a certain number of tokens. The second, Query-related Drop, drops tokens with the lowest scores based on their relevance to the overall query. The final strategy is our method, FoE-related Drop, which drops redundant tokens based on their relevance to the Focus-on-Emotion prompt. As shown in Tab. 2, FoE-related Drop outperforms the other two dropping strategies across different datasets, as this strategy focuses more on the relevance to emotion. In contrast, Query-related Drop may include more irrelevant information due to its broader textual reference. Notably, in the case of Random Drop, increasing the drop rate leads to more

inference inaccuracies, indicating that not all visual tokens are meaningless. Discarding useful information reduces inference accuracy, emphasizing the need for a well-chosen dropping strategy. In conclusion, choosing which visual tokens to drop based on their relevance to emotion is the most reasonable and effective approach (**RQ2**).

## 4.3. Diagnostic Analysis

We conduct ablation and visualization experiments on the Emotion6 and EmoSet datasets for in-depth analysis.

**Key Component Analysis.** We evaluate the effects of key components in SEPM, including Confidence-Guided Coarse-to-Fine Inference (CCI), Focus-on-Emotion Prompts (FP), and Visual Token Augmentation (VTA). As VTA depends on other components, it was not studied alone in ablation. The results are shown in Tab. 3, highlighting accuracy changes across components. Using a single component modestly improves inference accuracy. Moreover, combining CCI, FP, and VTA yields the best results. These findings show the key role of components in boosting performance.

**Sensitivity Analysis.** To demonstrate the robustness of SEPM, we conduct a sensitivity analysis of key hyperparameters. Focusing on overall model performance, we evaluate different settings for the loss parameters, confidence threshold $\alpha$, and drop rate $\beta$ on the Emotion6 and EmoSet datasets.

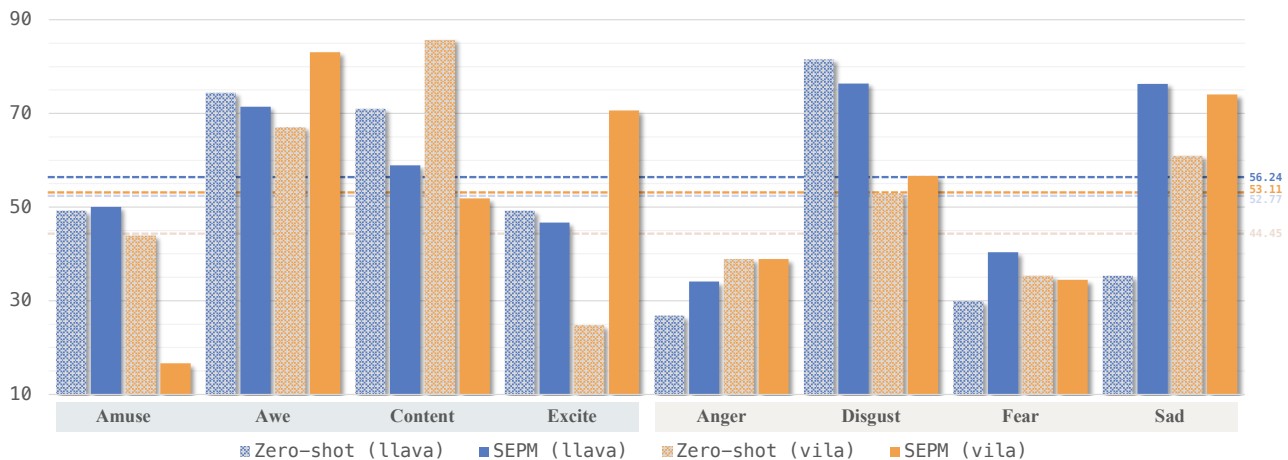

*Figure 6.* **Visualization** of fine-grained emotion classification on EmoSet. The left side is the positive emotion categories, and the right side is the negative ones. The dashed line represents the overall performance. Please see details in Sec. 4.3.

Specifically, we vary one hyperparameter at a time while keeping the others fixed. As shown in Fig. 3, model performance initially improves with increasing $\alpha$, then declines. A low threshold fails to distinguish reliable inferences in the first stage, while a high threshold misclassifies many unreliable predictions as reliable, weakening prompt enhancement in the second stage. Additionally, performance steadily declines as $\beta$ increases. A small drop rate reduces emotional redundancy and slightly boosts performance, but a high drop rate removes important visual information, degrading final results. Based on these observations, we select appropriate values for $\alpha$ and $\beta$ as the final settings.

**Visualization of Dropped Visual Tokens.** To demonstrate the reliability of visual token dropping, we visualize the dropping process of visual tokens in images across different emotions. In Fig. 4, each set of images from left to right consists of the original image, the image after a 20% drop rate, and the image after a 40% drop rate. As the drop rate increases, the amount of emotionally redundant information in the image decreases, with the remaining content primarily consisting of more significant emotional cues. This indicates that by selecting effective emotional cues based on the correlation between visual tokens and the Focus-on-Emotion textual prompt, we are able to retain and highlight the most relevant emotional information while discarding less important, redundant visual elements. It also reinforces that selecting visual tokens to drop based on their relevance to emotion is an appropriate approach (**RQ2**).

**Visualization of Confidence Estimation.** To validate the rationale of calculating the coarse-grained inference confidence based on the the logits distribution, we conduct an evaluation of inference accuracy at different variances the logits distribution in the first stage. As shown in Fig. 5, as the variance increases, the inference accuracy gradually improves, indicating that the closer the logits of different op-

tions are, the greater the likelihood of inference errors. This also means that the logits distribution effectively characterizes inference precision (**RQ3**). The confidence threshold we choose effectively avoids the variance intervals with poor accuracy, eliminates the interference from low-confidence regions, and enhances the overall performance of the model.

**Visualization of Fine-grained Emotion Classification.** To explore the preferences of MLLMs for different emotion categories, we calculate and visualize the accuracy of zero-shot and SEPM for each category in the EmoSet dataset under different architectures, as shown in Fig. 6. Our method improves overall performance, and we observe a striking phenomenon: the accuracy for each category fluctuates significantly. In semantically similar categories, an accuracy increase in one can cause a decrease in others. Since our method enhances prompts, we find that MLLMs exhibit different preferences based on the textual prompts (**RQ4**). Currently, we focus on improving overall performance, while addressing model preferences and selectively enhancing accuracy for each category is an area for further exploration.

## 5. Conclusion

In this paper, we address the challenges of emotion recognition in Multimodal Large Language Models (MLLMs), particularly the confusion caused by semantically similar emotion categories and visual redundancy. We propose Sharpening Emotion Perception in Multimodal Large Language Models (SEPM), which introduces a Confidence-Guided Coarse-to-Fine Inference framework and Focus-on-Emotion Visual Augmentation. These methods enhance emotion classification accuracy without additional training or manual annotation. Our approach, validated through experiments on various emotion datasets, demonstrates its effectiveness. We believe this work offers valuable insights for future research in MLLMs and emotion recognition.

**Acknowledge** This work is supported by the National Key Research and Development Program of China (2024YFC3308400), and National Natural Science Foundation of China under Grant (62272354, 62361166629, 62176188, 62225113, 623B2080). The supercomputing system at the Supercomputing Center of Wuhan University supported the numerical calculations in this paper.

## Impact Statement

This paper presents work whose goal is to advance the field of Machine Learning. There are many potential societal consequences of our work, none which we feel must be specifically highlighted here.

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
