# OpenReview forum: "Catch Your Emotion: Sharpening Emotion Perception in Multimodal Large Language Models"
_ICML.cc/2025/Conference — ICML 2025 spotlightposter_

### Official Review · Reviewer_W74m · 2025-03-03

**Overall Recommendation:** 5

**Summary:**

This paper proposes a method called Sharpening Emotion Perception in Multimodal Large Language Models (SEPM) to improve emotion recognition in MLLMs. It addresses two challenges: confusion between semantically similar emotions and the visual redundancy that distracts from emotional cues. SEPM incorporates a two-stage inference process and focuses on emotion-related visual cues, enhancing classification accuracy. The approach is training-free, requiring no additional manual annotations or fine-tuning, and significantly improves MLLM performance on emotion tasks, offering a resource-efficient and scalable solution for emotion recognition.

**Claims And Evidence:**

YES

**Essential References Not Discussed:**

None

**Experimental Designs Or Analyses:**

The experimental results are comprehensive. The method has been tested on different datasets, and the results are shown to be reliable through ablation studies and visualization experiments.

**Methods And Evaluation Criteria:**

Yes, the proposed method can effectively enhance the model's ability in sentiment reasoning.

**Other Comments Or Suggestions:**

The paper mentions that the model is highly sensitive to different prompts. I hope future work can explore this aspect further.

**Other Strengths And Weaknesses:**

Pros:
SEPM improves emotion recognition in MLLMs without requiring additional training or manual annotations, making it a resource-efficient and scalable solution for real-world applications.

Cons:

-	The Focus-on-Emotion prompt designed in the paper leads to significant performance improvements. I'm curious to know if this prompt is the only one that works, or if changing the prompt would have an impact on performance.

-	In theory, removing more visual information could mean losing important emotional cues needed for the model to make accurate predictions, but this is not clearly reflected in the paper's sensitivity analysis.

-	The description of the selection process based on the confidence threshold in the first stage is too simplistic in the paper.

-	The paper contains some inappropriate expressions.

**Questions For Authors:**

None

**Relation To Broader Scientific Literature:**

The author references the latest related work and proposes a method that takes a different approach compared to existing studies.

**Theoretical Claims:**

The approach is logically clear and the ideas are innovative.

---

> ### Author Rebuttal · Authors · 2025-03-29
>
> Dear Reviewer W74m:
>
> We are deeply grateful for your positive feedback on our work and the insightful suggestions. We have carefully reviewed each point and provided detailed responses accordingly.
>
> **Q1:  Performance under different prompts** (Other Strengths And Weaknesses)
>
> To explore the performance under different prompts, we designed a prompt augmentation semantically similar to Focus-on-Emotion: *Identify the predominant emotion conveyed*, and conducted corresponding experiments. As presented in the table, the performance of the new prompt is noticeably inferior to that of Focus-on-Emotion. We hypothesize that more direct prompts tend to yield better results. We plan to further explore the impact of various prompt designs on performance in future work.
>
> *Table: Ablation experiments on different prompts*
>
> |                Prompt type                | Drop rate = 0.1 | Drop rate = 0.2 |
> | :---------------------------------------: | :-------------: | :-------------: |
> | Identify the predominant emotion conveyed |      50.53      |      50.77      |
> |             Focus on emotion              |    **56.04**    |    **56.24**    |
>
> **Q2:  More analysis on removing visual information** (Other Strengths And Weaknesses)
>
> Compared to general tasks, visual redundancy in emotion-related tasks is more likely to interfere with the inference bias. Therefore, dropping a small amount of the least relevant visual information can help improve the model’s performance. In Figure 3 of the original paper, we conduct a sensitivity analysis under different drop rates and observe that discarding a small amount of redundant information leads to performance gains. However, excessive dropping results in the loss of some informative content, causing a decline in performance to some extent. Additionally, Figure 4 of the paper visualizes the specific content being dropped, showing that at lower drop rates, the removed tokens are mostly irrelevant, which allows the model to focus more on salient information and enhances its reasoning capability.
>
> **Q3:  The confidence-based prompt selection process in the first stage** (Other Strengths And Weaknesses)
>
> Sorry for the confusion. To encourage the model to focus on relatively simpler problems at each stage, we introduce prompt enhancement for fine-grained inference based on the outcomes of coarse-grained inference. To mitigate the risk of error propagation from the first stage adversely impacting the second-stage inference, we assess the confidence of the model using the variance of the output logits, rather than relying solely on predicted labels. Specifically, when the variance among logits is high (indicating strong confidence), we apply direct positive/negative prompt augmentation. Conversely, when the variance is low (suggesting uncertainty), we incorporate ambiguity-related descriptions to characterize the model decision process in the first stage, which are then used to guide the second stage. The selection of the variance threshold is discussed in the experimental section, where the effectiveness of the proposed strategy is demonstrated.
>
> **Q4:  Inappropriate expressions** (Other Strengths And Weaknesses)
>
> We will revise the inappropriate expressions in the final version. Thanks for your suggestions.

---

### Official Review · Reviewer_6rbK · 2025-03-06

**Overall Recommendation:** 4

**Summary:**

This paper proposes SEPM to tackle emotion recognition challenges in multimodal models. It focuses on issues like confusing similar emotions and visual noise. SEPM introduces a two-stage inference process: a coarse-to-fine approach to improve confidence in emotion classification and a focus on relevant emotional cues to reduce visual redundancy. The method improves performance without needing extra training, offering a scalable and efficient solution for emotion recognition tasks.

**Claims And Evidence:**

Yes.

**Essential References Not Discussed:**

NaN.

**Experimental Designs Or Analyses:**

The experimental results are comprehensive.

**Methods And Evaluation Criteria:**

Yes. The method is well evaluated.

**Other Comments Or Suggestions:**

None.

**Other Strengths And Weaknesses:**

Strengths：
1.The method is novel, and the experiments are comprehensive.
2.The paper is easy to understand.

Weaknesses：
1.The paper does not clearly specify how the attention map is obtained in the method, and it would be helpful to provide more details on this.
2.The experiments are relatively thorough, but the ablation study includes fewer metrics. It would be helpful to provide more information.
3.I would like to see an analysis of the inference efficiency.

**Questions For Authors:**

Refer to weaknesses.

**Relation To Broader Scientific Literature:**

The author introduces an innovative method that enhances the sentiment reasoning ability of MLLM through a two-stage inference process and visual augmentation.

**Theoretical Claims:**

The theory used in the paper is clear and easy to understand.

---

> ### Author Rebuttal · Authors · 2025-03-29
>
> Dear Reviewer 6rbK:
>
> Thank you very much for your valuable comments and constructive feedback. Below, we carefully respond to each of your concerns point-by-point, providing detailed explanations and supplementary evidence to further clarify our approach and demonstrate its effectiveness.
>
> **Q1:  Attention map computation** (Other Strengths And Weaknesses)
>
> Sorry for any inconvenience caused. Within the transformer architecture of MLLM, each block generates an attention map that reflects the model focus at that layer. Notably, attention maps from deeper layers tend to encode more abstract, high-level semantic representations, aligning more closely with the model learned conceptual understanding of the input. Therefore, we use the attention map from the final transformer block as the basis for token dropping in the second stage. Additionally, to enhance the accuracy of emotional focus, we retain only the attention maps between visual tokens and the Focus-on-Emotion prompt.
>
> **Q2:  More comprehensive ablation experiment** (Other Strengths And Weaknesses)
>
> Since we perform a classification task using an MLLM, the metric Accuracy serves as an appropriate indicator of performance. We conducted further ablation studies on the EmoSet dataset. As shown in the table, the effectiveness of each module is further demonstrated. More details will be provided in the final version.
>
> *Table: Ablation experiments on additional datasets*
>
> |     CCI      |      FP      |     VTA      | Emotion6 (ACC) | EmoSet (ACC) |
> | :----------: | :----------: | :----------: | :------------: | :----------: |
> |              |              |              |     48.32      |    52.77     |
> | $\checkmark$ |              |              |     51.68      |    53.98     |
> |              | $\checkmark$ |              |     51.52      |    54.28     |
> |              |              | $\checkmark$ |     53.03      |    56.10     |
> | $\checkmark$ | $\checkmark$ | $\checkmark$ |   **54.04**    |  **56.24**   |
>
> **Q3:  Discussion on inference efficiency** (Other Strengths And Weaknesses)
>
> Although the two-stage inference process introduces a certain increase in inference time, the overall inference latency for emotion recognition tasks remains low, due to the relatively lightweight nature of the task, making this additional overhead negligible in practical use. Additionally, we effectively reduced part of the inference time by discarding tokens. Overall, the slight extra time cost results in improved inference performance.

---

> > ### Comment · Reviewer_6rbK · 2025-04-07
> >
> > I think all the concerns have been addressed by the authors. I will maintain the score.

---

### Official Review · Reviewer_jWJJ · 2025-03-11

**Overall Recommendation:** 4

**Summary:**

This paper presents Sharpening Emotion Perception in MLLMs (SEPM), a training-free method to enhance emotional reasoning in multimodal large language models. SEPM improves emotion classification by using a Confidence-Guided Inference framework and Focus-on-Emotion Visual Augmentation to reduce distractions. Experimental results show significant performance improvements in emotion-related tasks, offering a scalable, resource-efficient solution.

**Claims And Evidence:**

Yes.

**Essential References Not Discussed:**

There is none.

**Experimental Designs Or Analyses:**

The experimental results are comprehensive. The author conducts various experiments with different relative method in various downstream tasks.

**Methods And Evaluation Criteria:**

Yes. The method is relatively applicable.

**Other Comments Or Suggestions:**

It is suggested to correct the minor errors in the paper.

**Other Strengths And Weaknesses:**

Pros:
• The paper is well-structured and clearly written.
• SEPM is able to enhance emotion recognition in MLLMs without requiring additional training or manual annotations.

Cons:
• The generalizability of SEPM is not fully established, as its performance and suitability for other multimodal tasks have not been adequately validated, indicating the need for additional experiments and data validation.
• Why does the acc in Fig. 3 show little change with \beta in on the EmoSet dataset? Additionally, how do larger \beta values affect the results? Further discussion is needed.
• The meaning of Eq. (6) is unclear; further explanation would be appreciated.
• The numbers on the right side of the dashed line in Fig. 6 are too small, resulting in poor readability.

**Questions For Authors:**

None.

**Relation To Broader Scientific Literature:**

Based on the issues in existing work, the author proposes a new approach.

**Theoretical Claims:**

The paper presents a simple and easy-to-understand approach, but provides limited discussion on the theory.

---

> ### Author Rebuttal · Authors · 2025-03-29
>
> Dear Reviewer jWJJ:
>
> Thank you again for your thoughtful and constructive suggestions. In the following responses, we address each of your points thoroughly, providing additional explanations and supporting evidence to strengthen the clarity of our methods.
>
> **Q1: Discussion on generalizability of SEPM** (Other Strengths And Weaknesses)
>
> Thank you for highlighting this issue. It should be clarified that 1) the two-stage inference process in our method is specifically designed to address the progressive categorization from coarse-grained to fine-grained levels inherent in emotion recognition tasks, and 2) the visual enhancement component of our method is motivated by the interference caused by redundant visual information in emotion-related tasks. Consequently, both modules are specifically tailored for emotion recognition and are not directly transferable to other tasks. Nonetheless, we have explored applying the Focus-on-X prompt to other domains and conducted experiments on a medical dataset (VQA-RAD [1]). As shown in the table, models across various architectures all exhibited improved performance, indicating that Focus-on-X can effectively serve as a generalized prompt enhancement.
>
> *Table: Generalizability experiment on the VQA-RAD Dataset (Medical Domain)*
>
> |     Dataset      | LLaVA-7b  |  VILA-8b  |
> | :--------------: | :-------: | :-------: |
> |    Zero-shot     |   37.47   |   38.58   |
> | Focus-on-Medical | **39.25** | **40.58** |
>
> **Q2: Experiment on higher drop rate** (Other Strengths And Weaknesses)
>
> We conducted ablation experiments with higher drop rates (beta), as shown in the table below. As the drop rate increases, the performance of the model declines to some extent but still maintains a certain inference capability. We believe discarding the least relevant redundant information helps the model better focus on emotion-related content, thereby improving performance. However, as the drop rate continues to increase, the model begins discarding some moderately important information, leading to a reduction in inference accuracy. Nonetheless, since the most critical emotional information is preserved, the model's performance does not completely collapse.
>
> *Table: Ablation experiment on drop rate*
>
> | Drop rate |  0.1  |    0.2    |  0.3  |  0.4  |  0.5  |  0.6  |  0.7  |  0.8  |
> | :-------: | :---: | :-------: | :---: | :---: | :---: | :---: | :---: | :---: |
> |    ACC    | 56.04 | **56.24** | 55.97 | 55.96 | 53.95 | 53.82 | 53.81 | 52.51 |
>
> **Q3: Lack of clarity in Equation 6** (Other Strengths And Weaknesses)
>
> Sorry for the confusion. We have revised the relevant part of Equation 6:
>
> Then we extract the FoE prompt set as emotion-related text, leveraging the query-dimension of textual logits and the key-dimension of the visual modality to construct the drop matrix $P \in \mathbb{R}^{L_t \times L_v}$, where $L_t$ and $L_v$ are the lengths of the text tokens (FoE) and visual tokens, respectively. The drop matrix $P$ is defined as:
> $$
> P[i, j] = \mathcal{A}(i, j)
> $$
> $i \in$ { $x \mid \mathbb{I}[x] \in \mathbb{T}$ },  $\quad j \in$ { $y \mid \mathbb{I}[y] \in \mathbb{V}$ },
>
> where $\mathbb{I}$ denotes the set of all input tokens, including both textual and visual modalities, $\mathbb{T}$ denotes FoE prompt token set and $\mathbb{V}$ is visual tokens set, i.e.
> $$
> (\mathbb{T} \cup \mathbb{V}) \subseteq \mathbb{I}
> $$
> We will update the above content in the final version.
>
> **Q4: Problem on Figure 6** (Other Strengths And Weaknesses)
>
> Thank you for tips. We will update in the final version.
>
> [1] A dataset of clinically generated visual questions and answers about radiology images, Scientific data, 2018

---

> > ### Comment · Reviewer_jWJJ · 2025-04-03
> >
> > The authors have addressed most of the concerns, and the supplement to the methods and experiments is clear. I am inclined to accept.

---

### Official Review · Reviewer_wn8T · 2025-03-15

**Overall Recommendation:** 4

**Summary:**

This paper proposes a training-free approach for emotion classification using Multimodal Large Language Models (MLLMs). They find that MLLMs (1) struggles to distinguish between semantically similar emotions, and (2) are overwhelmed by redundant visual information. To address these challenges, they propose a Coarse-to-Fine inference framework to refine emotion classification and a Focus-on-Emotion Viusal Augmentation approach to reduce visual redundancy. Experimental results on multiple benckmarks demonstrate the effectiveness of the proposed method.

## update after rebuttal

I read authors' response and other reviewers' comments. I appreciate the clarifications and new results, which have fully addressed my concerns. Therefore, I will raise my score.

**Claims And Evidence:**

yes

**Essential References Not Discussed:**

no

**Experimental Designs Or Analyses:**

no

**Methods And Evaluation Criteria:**

yes

**Other Comments Or Suggestions:**

1. I recommend the authors further improve the methodology and highlight the innovations.

2. More methods could be included for comparison to strengthen the comprehensiveness of the evaluation.

**Other Strengths And Weaknesses:**

strengths
1. This paper propose a training-free approach for zero-shot emotion perception using MLLMs, providing an effective and efficient solution in this area.

2. Experimental results on multiple benchmarks and extensive diagnostic analysis demonstrate the effectiveness.

3. The writing is accessible, making the paper easy to follow.

weaknesses


1. The innovations of this paper is concerning. In particular, the proposed Focus-on-Emotion Visual Augmentation is methodologically idential to the adaptive sparsification in [**SPARSEVLM'25**](http://export.arxiv.org/pdf/2410.04417), which employ the same approach to drop irrelavant visual tokens.

2. The author does not compare their method to those in the emotion-related field (e.g., [**SoV'24**](https://arxiv.org/pdf/2410.02244), which is also zero-shot), making the evaluation less convinced. At least, the rationale behind this choice should be clarified.

**Questions For Authors:**

see weaknesses

**Relation To Broader Scientific Literature:**

This paper propose a training free approach for emotion perception using MLLMs, offering a new well-performed solution. However, one of the core contributions of this paper, i.e., Focus-on-Emotion Visual Augmentation, as the author claimed,  has been proposed in prior work ([**SPARSEVLM'25**](http://export.arxiv.org/pdf/2410.04417)), which focus on more common VQA tasks. Although they achieves best overall results in emotion perception task, the contributions of this paper are less significant to the community under a broader background.

**Theoretical Claims:**

no

---

> ### Author Rebuttal · Authors · 2025-03-29
>
> Dear Reviewer wn8T:
>
> We sincerely appreciate your time and effort in reviewing our paper. We have provided further clarification on each of the issues you raised. We hope the detailed responses below fully address your concerns, and we would be grateful if you would consider updating your score.
>
> **Q1: Different with SPARSEVLM & innovations** (Relation To Broader Scientific Literature & Other Strengths And Weaknesses & Other Comments Or Suggestions)
>
> Thank you for raising this valuable question. Compared to general tasks, emotion recognition is often susceptible to interference from redundant visual information (noise), leading to inference biases. While SparseVLM accelerates inference for general tasks by retaining only the most crucial information in the visual modality, our method, in contrast, designs a specialized Focus-on-Emotion prompt, effectively identifying emotion-related visual redundancy through prompt-guided relevance. By discarding such redundant information, we can significantly enhance the accuracy of the model in emotion inference. Additionally, in emotion recognition tasks with a large number of categories, semantic similarities between emotions often lead to confusion. Therefore, we introduce Confidence-Guided Coarse-to-Fine Inference, enabling the model to handle simpler subtasks incrementally, thereby improving overall performance.
>
> **Q2: Reason for not comparing with other emotion-related methods** (Other Strengths And Weaknesses)
>
> To the best of our knowledge, we are the **first** to explore training-free optimization in general emotion recognition tasks. Therefore, we do not identify suitable emotion-related methods for a fair and meaningful comparison. Although the SoV [1] method mentioned in the paper is also training-free, it is specifically designed for facial emotion images and requires additional manual bounding box annotations. Moreover, since its code has not been released, we do not have the capacity to adapt it to our model architecture and more general emotion recognition tasks in a short period of time. We will try to include a comparison with SoV in the final version, and will continue to follow similar work in the future. Thank you for your valuable suggestion.
>
> **Q3: More comparative experiment** (Other Comments Or Suggestions)
>
> In the absence of suitable emotion-related baselines for direct comparison,  we introduced a comparative experiment with PDrop [2] based on its official code, which aims to eliminate visual redundancy to optimize the model. The experimental results in the table show that PDrop yields a slight improvement over the zero-shot setting by mitigating visual redundancy, supporting the hypothesis that emotional visual redundancy may interfere with the judgment of the model. Moreover, the results highlight the superior effectiveness of our proposed method.
>
> *Table: Comparison with PDrop on various emotion datasets.*
>
> |   Dataset   | Emotion6  |  EmoSet   | WebEmo7 | WebEmo25 | Abstract  |  Average  |
> | :---------: | :-------: | :-------: | :--------: | :---------: | :-------: | :-------: |
> |    LLaVA    |   48.32   |   52.77   |   25.56    |    15.71    |   27.86   |   34.04   |
> |    PDrop    |   51.24   |   53.53   |   27.50    |    16.02    |   27.50   |   35.16   |
> | SEPM (ours) | **54.21** | **56.04** | **42.39**  |  **18.26**  | **29.29** | **40.04** |
>
> **Thank you for your valuable advice. We are deeply grateful and appreciate your guidance.**
>
> [1] Visual Prompting in LLMs for Enhancing Emotion Recognition, EMNLP, 2024
>
> [2] PyramidDrop: Accelerating Your Large Vision-Language Models via Pyramid Visual Redundancy Reduction, CVPR, 2025

---

### Decision · Program_Chairs · 2025-05-01

**Decision:**

Accept (spotlight poster)

**Comment:**

This paper introduces a training-free framework to enhance emotion recognition in MLLMs. Empirical evaluations on multiple emotion recognition benchmarks show that SEPM offers notable improvements over baseline zero-shot settings and some alternative methods (such as PDrop). The authors also include an exploratory study showing potential generalizability to other domains (e.g., medical VQA), though the main focus remains on emotion recognition. There is some disagreement among the reviewers. One reviewer expressed reservations (leaning towards weak rejection) mainly due to methodological overlap with existing work and limited baseline comparisons. However, the remaining reviewers lean toward acceptance, with scores ranging from Accept to Strong Accept.

Given the strong empirical results, clear scalability and efficiency benefits, and the authors’ responsiveness in addressing the core concerns, the overall recommendation is to accept the paper, with the expectation that the final version will incorporate the clarifications and additional comparisons where possible.